# Social Commonsense-Guided Search Query Generation for Open-Domain Knowledge-Powered Conversations

**Revanth Gangi Reddy  Hao Bai  Wentao Yao  Sharath Chandra Etagi Suresh**
**Heng Ji  ChengXiang Zhai**
University of Illinois at Urbana-Champaign
{revanth3, haob2, wentaoy4, sce3, hengji, czhai}@illinois.edu

## Abstract

Open-domain dialog involves generating search queries that help obtain relevant knowledge for holding informative conversations. However, it can be challenging to determine what information to retrieve when the user is passive and does not express a clear need or request. To tackle this issue, we present a novel approach that focuses on generating internet search queries that are guided by *social commonsense*. Specifically, we leverage a commonsense dialog system to establish connections related to the conversation topic, which subsequently guides our query generation. Our proposed framework addresses passive user interactions by integrating topic tracking, commonsense response generation and instruction-driven query generation. Through extensive evaluations, we show that our approach[1] overcomes limitations of existing query generation techniques that rely solely on explicit dialog information, and produces search queries that are more relevant, specific, and compelling, ultimately resulting in more engaging responses.

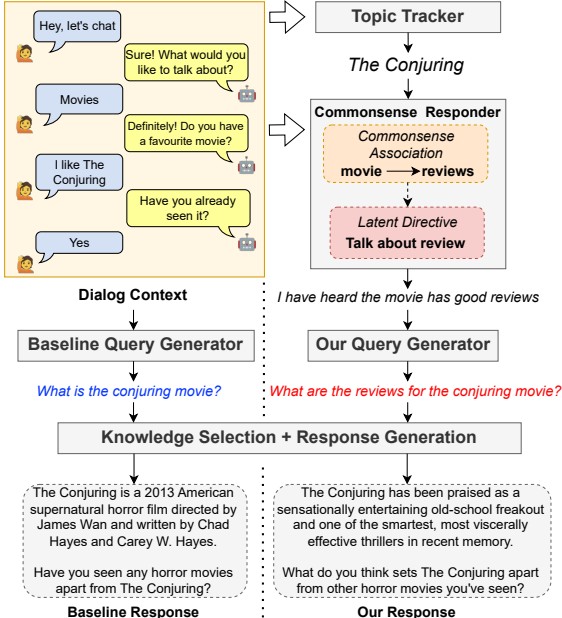

Figure 1: Our proposed query generation vs a baseline query generator. The commonsense responder identifies associations related to the dialog topic (e.g., *movies* → *reviews*) and provides a latent directive (in the form of a response) to guide the generation of search queries.

## 1 Introduction

Conversational systems have evolved to include personal assistants, task-oriented bots, and open-domain dialog agents for casual conversations. For these agents to maintain engaging and informative discussions, it is crucial to access external knowledge. Holding a knowledge-powered dialog (Dinan et al.; Komeili et al., 2022; Li et al., 2022; Lai et al., 2023a) typically involves the generation of a search query that can help gather the most relevant information to continue the conversation. While such queries are more obvious when the user explicitly asks for certain information, a.k.a. conversational information seeking (Zamani et al., 2022), it is unclear what information should be pursued when users are passive, disengaged, and do not provide

clear guidance for the conversation (Hardy et al., 2021). Nevertheless, in open-domain conversations, users can introduce any topic, and designing a comprehensive algorithm that can produce a relevant query in response to a random user topic poses a unique, complex challenge that has not been explored in prior research.

To tackle this challenge, we propose to integrate social commonsense reasoning for the generation of search queries in knowledge-powered conversations. Social commonsense (Moore, 2006) refers to the general knowledge about social situations and human behavior used to connect conversation topics and guide discussions. We thus hypothesize that by leveraging a deeper understanding of social commonsense and the unspoken cues that guide human conversation, chatbots can become more

---

[1]Code and models are available here: 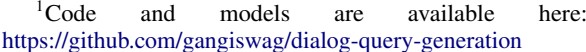
https://github.com/gangiswag/dialog-query-generation

adept at navigating passive conversations.

Concretely, we introduce a novel framework that uses a commonsense response as a *latent directive* for an instruction-following query generator. Our approach incorporates the use of topic tracking (§2.1) to first identify the main point of discussion, followed by commonsense-based response generation that can associate concepts to the main topic to give a latent commonsense directive (§2.2). Finally, we use instruction-driven query generation (§2.3) to output a search query that adheres to the latent directive within the commonsense response.

Our method overcomes the limitations of existing techniques (Shuster et al., 2022a,b; Cai et al., 2022; Lai et al., 2023b) that solely depend on explicit information present in the conversation to generate search queries. Such an approach is suboptimal in case of passive conversations, where the human isn't necessarily asking for any specific information. Figure 1 shows an example comparing our approach against a baseline query generation system (Shuster et al., 2022b). Our topic tracking identifies 'The Conjuring' as the subject, with the commonsense responder making the association *movie → reviews* to output a latent commonsense directive that refers to discuss movie reviews. This directive guides the search query generator to output a query for the movie reviews, the results of which lead to a more engaging bot response compared to the baseline, as can be seen in Figure 1.

## 2 A Novel Query Generation Framework

In this section, we present our framework for generating search queries by leveraging commonsense reasoning. Our approach consists of three main components: topic tracking to pinpoint the core subject, commonsense-based response generation that relates concepts with the primary topic and provides a latent commonsense directive, and instruction-driven query generation to produce a search query capable of retrieving relevant information that follows the commonsense directive. Figure 2 illustrates how these components are integrated, with each step described in detail below.

### 2.1 Fine-Grained Topic Tracking

Topic tracking (Nakata et al., 2002) aims to identify the primary subject of the discussion in free-form dialogs, and has been demonstrated (Guo et al., 2018) to improve the coherence of dialog systems. Unlike previous approaches (Khatri et al., 2018)

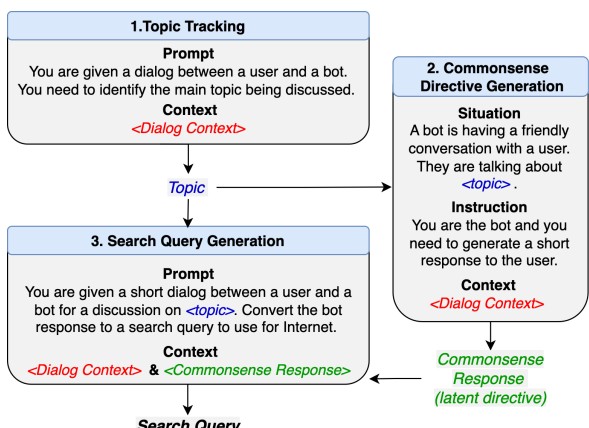

Figure 2: Interaction of different components within our proposed query generation framework.

that track a fixed set of broad high-level topics (e.g., *movies*, *sport*), our objective is to detect unconstrained, finer-grained topics (such as *movie/actor* names or *teams*). For fine-grained topic tracking, we apply an instruction-tuned model (Chung et al., 2022) to identify the current topic from the dialog context. We utilize the prompt in Figure 2 and rely on instruction-tuned models with strong zero-shot abilities (Wei et al., 2021) due to lack of training data for such topic tracking in dialog[2]. An alternative topic tracking approach could follow Shuster et al. (2022a); Adolphs et al. (2022), extracting topics as relevant entities grounding the final response.

### 2.2 Commonsense-Based Directive

Social commonsense-based dialog systems (Kim et al., 2022a,b; Zhou et al., 2021) typically demonstrate a fundamental understanding of handling and responding to specific topics or situations. They involve using external commonsense knowledge graphs such as ConceptNet (Speer et al., 2017) or ATOMIC (Sap et al., 2019) to collect triples for response generation (Zhou et al., 2022), or distilling such knowledge into language models (LM) through large-scale pretraining (Kim et al., 2022a; Chen et al., 2023) for direct response generation. In this work, we adopt the latter approach, by using a pretrained LM to derive the commonsense directive in the form of a response.

Specifically, we use Cosmo (Kim et al., 2022a), which was trained on socially-grounded synthetic dialogues generated by prompting InstructGPT (Ouyang et al., 2022) using contextualized commonsense knowledge from ATOMIC. Cosmo takes

---

[2]We note that topic tracking is related to open-domain dialog, and is different from state tracking (Williams et al., 2013) which is specific to task-oriented dialog.

a situation narrative and role instruction as input, and generates a response based on the dialog context. We also integrate the topic tracking output into the situation narrative definition, as illustrated in Figure 2. Subsequently, Cosmo's output serves as the latent commonsense directive to guide search query generation, which is discussed next.

## 2.3 Instruction-Driven Query Generation

Given the dialog context, conversation topic and a latent directive in the form of a commonsense response, we aim to generate a search query to obtain relevant information for continuing the conversation. We utilize an instruction-tuned model (Chung et al., 2022) for query generation, by prompting (see Figure 2) it to transform the commonsense response into a search query, while incorporating the fine-grained topic to enhance relevance and specificity. Essentially, the commonsense response embodies the bot's informational requirement, guiding it to obtain the mentioned information.

## 3 Experiments

### 3.1 Setup

**Dataset**   We use the Wizard of Internet (WoI) (Komeili et al., 2022) dataset for our experiments. WoI is a human-human dialog corpus for knowledge-powered conversations, with one of the speakers having internet access to gather information for generating responses.

**Models and Baselines**   The topic-tracker is based on Flan-T5 large (770M) (Chung et al., 2022) while the commonsense response generation uses the 3B version of Cosmo (Kim et al., 2022a). The query generator is also based on the Flan-T5 large model. We compare our query generation approach primarily against Blender Bot 3 (Shuster et al., 2022b), a state of the art open-domain conversational agent. We also compare against a version of our approach that does not incorporate the Cosmo response for query generation, termed *Flan T5 w/o Cosmo*.

**Finetuning with ChatGPT Annotations**   Our approach uses an instruction-tuned Flan T5 model in a zero-shot setting for topic tracking and query generation. To improve performance, we separately finetune the topic-tracker and query generator using ChatGPT annotations (the same prompt as Flan T5 is used to obtain silver labels from ChatGPT). To create finetuning data, we choose turns corresponding to internet search from the WoI training

set, yielding 20k examples.

**Internet Search and Response Generation**   We obtain search results by scoring passages from the top-3 Bing Search pages using a reranker[3]. With our primary focus being query generation, we simply prompt ChatGPT to generate the response by incorporating the top search result given the dialog context. We also consider a "no query" baseline that corresponds to generating responses directly from ChatGPT (*gpt-3.5-turbo-0301* version) without internet search. For both search query generation and final response generation, we employ the nucleus sampling method (Holtzman et al., 2019) with $P$ set at 0.9 and a temperature of 0.7. We set max tokens to 40 and 100 for search query generation and final response generation respectively.

### 3.2 Evaluation

In our evaluation, we focus on WoI test set with dialog turns that had search queries annotated for generating responses, while specifically targeting "passive turns" where users don't explicitly request information. Using an intent detection model (Khatri et al., 2018), we identify and remove turns related to information or opinion requests, and randomly selected 200 examples for human evaluation.

**Human Evaluation**   We conducted a human study with four experienced NLP students to evaluate the quality of generated search queries and responses. Queries were assessed based on relevance, specificity, usefulness, and potential to maintain user engagement in the dialog. Responses were evaluated for engagement, coherence, and informativeness. Detailed guidelines are in the appendix.

**Automatic Evaluation**   Recent studies, like G-EVAL (Liu et al., 2023) and GPTScore (Fu et al., 2023), show that LLMs such as GPT-4 can effectively evaluate natural language generations and align well with human assessments. Therefore, we utilize GPT-4 for automatic evaluation, prompting[4] it to provide an overall score (ranging from 1-10) for search queries and final responses. As seen in §3.3, our human study results corroborate GPT-4 evaluations. Additionally, we use a ranker model (Hedayatnia et al., 2022) trained on the Alexa Prize Socialbot Grand Challenge (Johnston et al., 2023) response selection data (Ram et al., 2018) for response evaluation.

---

[3]We use the `ms-marco-MiniLM-L-6-v2` model.
[4]Detailed GPT-4 prompts are in the appendix.

| Query Generation Approach | Search Query | | | | | Final Response | | | | |
|---|---|---|---|---|---|---|---|---|---|---|
| | Human | | | | Automatic | Human | | | Automatic | |
| | Rel. | Spe. | Use. | Int. | GPT-4 | Eng. | Info. | Coh. | Ranker | GPT-4 |
| No Query | - | - | - | - | - | 2.71 | 1.87 | **3.11** | 78.9 | 66.2 |
| Blender Bot 3 | 3.13 | 2.29 | 2.61 | 2.28 | 35.1 | 2.85 | 3.19 | 2.88 | 75.4 | 68.0 |
| Flan T5 w/o Cosmo | 3.38 | 3.21 | 3.06 | 3.02 | 44.3 | 3.01 | 3.27 | 2.92 | 76.9 | 67.6 |
| Ours (Zero-shot) | 3.59* | 3.51* | 3.39* | 3.29* | 49.9* | 3.13 | **3.35** | 3.00 | 78.6 | 70.6* |
| Ours (Finetuned) | **4.16*** | **4.05*** | **3.98*** | **3.91*** | **72.2*** | **3.29** | 3.31 | **3.10** | **80.7** | **72.1** |
| ChatGPT | 4.51 | 4.49 | 4.48 | 4.45 | 80.7 | - | - | - | - | - |

Table 1: Evaluation of different query generation approaches on the WoI dataset, based on the quality of search queries and final responses. For query generation (left), *Finetuned* refers to leveraging dataset dialogs, while *zero-shot* corresponds to instruction-tuned. For response generation (right), responses from ChatGPT are conditioned on internet search results obtained using the corresponding queries. The acronyms for human evaluation are: **Rel**evance, **Spe**cificity, **Use**fulness, **Int**erestingness, **Eng**agement, **Info**rmativeness, and **Coh**erence. * corresponds to statistical significance ($|z| > 3.3$ and $p < 0.05$).

## 3.3 Results

**Quality of generated search query**   Table 1 (left) shows the results of human and automatic evaluation of search queries. Mainly, we notice that instruction-tuned models outperform Blender Bot 3 significantly, and using Cosmo's commonsense response as a directive for guiding query generation with Flan T5 shows consistent improvements. Lastly, substantial enhancements in query quality are observed upon fine-tuning the zero-shot system with ChatGPT annotations. By computing the Spearman correlation between automatic metrics (GPT-4) and overall score for human evaluations (average of four aspect ratings), we found a strong correlation (0.674) between the two measures.

**Quality of final responses**   Table 1 (right) shows results for evaluation of the generated responses. We see that directly generating a response from ChatGPT without internet search can still lead to a very coherent response, but is less engaging and very uninformative. Our proposed query generation framework leads to consistent improvements across all aspects of the final response, particularly with high engagement scores. Notably, boosting engagement, or the likelihood of continued human-bot interaction, is crucial in passive conversations.

## 3.4 Analysis

**Instruction-Following Capability**   We study the impact of the query generator's instruction-following capability on utilizing the commonsense directive (i.e., cosmo output) to generate better queries. Using GPT-4 preference evaluation, we explore how increasing the query generator's size[5]

[5]As per Chung et al. (2022), larger instruction-tuned models usually have better instruction-following capability.

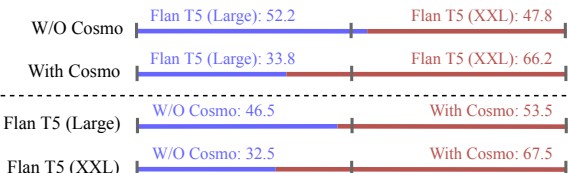

Figure 3: Examining the impact of a larger model for query generation, as evaluated by GPT-4's preference.

| | Relevant | Specific | Overall |
|---|---|---|---|
| W/O Topic Tracker | 43.0% | 39.5% | 39.0% |
| With Topic Tracker | **57.0%** | **60.5%** | **61.0%** |

Table 2: Aspect-wise vs overall relative preference scores, based on GPT-4 evaluation, for search queries generated with and without the topic tracking component in our framework.

influences the quality of generated queries both with and without the commonsense directive. Figure 3 reveals that incorporating the commonsense directive (a) significantly enhances query quality as model size increases, and (b) leads to greater improvement for larger models (67.5% for XXL vs 53.5% for Large). Hence, a more robust instruction-tuned model effectively leverages the commonsense directive in generating better queries.

**Benefit of Topic Tracking**   Within our framework, topic tracking serves to (a) maintain coherence between the generated query and the most recent discussion subject, and (b) assist in shaping Cosmo's situation narrative (refer to prompts in figure 2). Here, we study the benefit of topic tracking by removing it from the Cosmo and query generator inputs and evaluating the final query quality. Using GPT-4 for automatic preference evaluation, we compare queries generated with and without the

topic tracker. Table 2 shows results from the study, with GPT-4 finding queries generated by involving topic tracking better, particularly for relevance and specificity.

**Error Categorization** We examined 50 low-scoring examples[6] from the human evaluation of search queries produced by the zero-shot approach. The main error categories were: (i) *Incorrect Topic* (31.4%) - topic tracker failed to identify the current discussion subject, (ii) *Trivial Query* (29.4%) - query was obvious or already answered in the conversation history, (iii) *Query Instruction Mismatch* (23.5%) - query generator misunderstands instructions, and outputs conversational questions instead, and (iv) other irrelevant queries (15.7%). After finetuning with ChatGPT annotations, 70.6% of the queries significantly improved, while the rest maintained a more or less similar quality level. The breakdown of examples that continue to score low after finetuning is as follows: 19% result from *Incorrect Topic*, 41% from *Trivial Queries*, 5% from *Query Instruction Mismatch*, and 35% from other unrelated queries. Notably, finetuning effectively reduces 67% of *Trivial Query* errors, 75% of *Incorrect Topic* errors, and over 90% of *Query Instruction Mismatch* errors. This suggests that finetuning enhances topic tracking capabilities (resulting in fewer *Incorrect Topic* errors) and ensures better adherence to search query generation instructions (leading to a decrease in *Query Instruction Mismatch* errors).

## 4 Conclusion and Future Work

We introduce a novel framework that leverages commonsense to enhance the generation of search queries in internet-powered dialog. Our results show that incorporating a commonsense-based directive yields search queries with improved relevance, specificity and appeal, fostering user engagement in otherwise passive conversations. Future work will use more intricate social narratives by incorporating user preferences from past conversations, to align commonsense directives towards individual interests.

## Acknowledgement

We would like to thank Amazon Inc. for providing a grant partially supporting the work of the team. We thank the Amazon Alexa Prize team for their great support, guidance, and help. We are also grateful to the anonymous reviewers, the Blender NLP group and CharmBana team members for their valuable feedback and comments.

## Limitations

We expect the following limitations for our approach:

- **Focusing only on turns involving search**: Our methodology primarily targets the generation of search queries, hence we chose only those turns from the WoI dataset that contained annotated search queries. A more pragmatic approach would also require a search decision module to determine when it is essential to seek external information.

- **Assuming topic continuity in discussion**: Our method assumes a continuous presence of a discussion topic. Nevertheless, situations like topic shifts can cause temporary absence of a topic, which our approach does not consider. For instance, when the human suggests, "*Let's discuss something else*," there is no current topic for the discussion.

## Ethical Considerations

We raise the following ethical concerns from leveraging internet search for open-domain dialog:

- **Toxicity of retrieved content**: There is a need for a toxicity filter or a content moderation system to ensure that the retrieved content is safe, non-offensive, and free from any form of hate speech, discrimination, or harassment.

- **Privacy concerns**: Using Internet search for dialog systems can raise privacy concerns, as users might be discussing or sharing personal or sensitive information in their conversations. It is crucial to implement proper data anonymization and encryption techniques to protect users' privacy .

- **Reliability and credibility of sources**: Dialog systems must be cautious when referring to information from the Internet, as the content may not always be accurate or reliable. Verifying the credibility of sources and cross-referencing is essential to ensure that the information provided by the system is reliable and trustworthy.

---

[6]Qualitative examples are provided in the appendix

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

# A Experimental Setup

## A.1 Wizard of Internet Dataset

The Wizard of Internet (WoI) dataset (Komeili et al., 2022) comprises a vast collection of external knowledge-based conversations, allowing agents to leverage internet search to obtain relevant information. We utilize a training subset of WoI consisting of 6,029 conversations with 29,371 turns. Upon transforming the dataset, models are employed to sequentially generate three items for each bot utterance with human annotated search queries: the ChatGPT topic tracking output, commonsense directive from Cosmo, and the ChatGPT search query. From this, 20k instances are sampled to form the finetuning data.

## A.2 Models and Baselines

**Flan T5** (Chung et al., 2022) We employ the Flan T5 (Large, 770M) as the model for entity tracking and query generation. Given the context, we incorporate the prompt depicted in Figure 2 into the Flan T5 input using the entity and query created by ChatGPT as the ground truth for finetuning.

**Cosmo** (Kim et al., 2022a) We use the 3B variant of Cosmo model as our social commonsense-based response generator. The model has been trained on 1.5 million synthetically generated social dialogs from SODA (Kim et al., 2022a). Cosmo has been shown to be capable of managing conversations across diverse dialog situations.

**Blender Bot 3** (Shuster et al., 2022b) Blender Bot 3 (BB3) is a unified sequence-to-sequence language model designed for modular dialogue systems, which appends instruction tokens to the generated text. BB3 employs R2C2 (Shuster et al., 2022b) as the language model to create queries, consisting of a 2.7 billion-parameter Transformer architecture with 22 encoder and 22 decoder layers, pretrained on pushshift.io Reddit and RoBERTa + CC100en data (Shuster et al., 2022a). Furthermore, BB3 finetunes R2C2 on a variety of query datasets, such as the Wizard of Internet and Feedback on Interactive Talk & Search (FITS) (Xu et al., 2022).

## A.3 Qualitative Examples

Here, we show some qualitative examples for the search queries generated by different approaches, and the corresponding responses generated by incorporating results obtained using the query. Table 3 shows an example for each error type.

**ChatGPT Response Generation**

You are given a dialog between a bot and a user:

> **Bot:** ...
> **User:** ...
> ......
> **Bot:** ...
> **User:** ...

Generate a single-sentence response from the bot that incorporates the most relevant part from the content below.

Content:

    **<Search Content>**

**Bot:**

Figure 4: ChatGPT prompt for generating the bot response by incorporating search content.

---

**Dialog Context**

*Bot:* Seventeen always does great things for promotions on social media!
*User:* Yes they do, Their music is the best, Their dance chorography are even better!

*Gold Bot response:* They add an emotional tone to the chorography, which is awesome.

**Direct ChatGPT**
*Response:* Glad to hear that you enjoy both their music and dance performances!

**Blender Bot 3**
*Query:* seventeen
*Response:* Yes, Seventeen's name reflects the unique way they are structured as a group with 13 members divided into three units.

**Flan T5 without Cosmo (Zero-shot)**
*Query:* what is the best song on seventeen?
*Response:* Have you voted for your favorite Seventeen song on the list ranking the best of all time?

**Flan T5 with Cosmo (Zero-shot)**
*Query:* What is Seventeen's most popular song?
*Response:* Have you voted for your favorite Seventeen song yet? Click the link to make your voice heard in the ranking of their greatest hits.

**Flan T5 with Cosmo (ChatGPT Finetuned)**
*Query:* "What are some of Seventeen's most popular social media promotions?"
*Response:* It's great to hear that you appreciate Seventeen's social media presence, have you seen any of our favorite campaign examples from 2022 such as KFC's #UnboringMornings or Reddit's Super Bowl Awareness Campaign?

---

**Dialog Context**

*Bot:* I have heard that carrots probably originated in Persia.
*User:* I didn't know they originated in Persia. I have always hated them so I didn't care where they were from.

*Gold Bot response:* What is your favorite skyscraper to visit?

**Direct ChatGPT**
*Response:* That's interesting to know, even if you don't like them.

**Blender Bot 3**
*Query:* carrots
*Response:* Despite your dislike for them, carrots are actually a highly nutritious and beneficial root vegetable.

**Flan T5 without Cosmo (Zero-shot)**
*Query:* where did the origin of carrots come from?
*Response:* Yes, carrots were first cultivated in parts of Asia and the Middle East and were brought to Europe by the Arabs in the 10th century.

**Flan T5 with Cosmo (Zero-shot)**
*Query:* What are the different theories about the origin of carrots?
*Response:* Carrots have a single origin in Central Asia, but their wild ancestors probably originated in Persia and were selectively bred over the centuries to produce the familiar garden vegetable.

**Flan T5 with Cosmo (ChatGPT Finetuned)**
*Query:* What are some theories about where carrots originated in Persia?
*Response:* Yes, carrots actually have a single origin in Central Asia and their wild ancestors probably originated in Persia, which is still the centre of diversity for the wild carrot.

| Error Type (Ratio) | Examples |
|---|---|
| Entity **Incorrect Topic** (31.4%) | **Context**
*Bot*: In flames is such an interesting swedish band.
*User*: Yeah they have quite a few albums I think.
*Bot*: yeah they have 12 studio albums!
*User*: That's awesome. I'd love to find other bands that sound like them.
*Bot*: Wormrot is a metal band that is popular
*User*: Thanks, I'll be sure to check them out, I love going to live music shows, not just metal.
**Cosmo Response**
*Bot*: Yeah, live music is always a great experience.
**Zero-shot T5 with Cosmo**
*Entity:* Rock band
*Query:* What band is Wormrot?
*Problem:* Entity tracker is too general and query does not track a topic transition
**Finetuned T5 with Cosmo**
*Entity:* Wormrot (metal band)
*Query:* What are some notable live performances by Wormrot? |
| Query **Trivial** (29.4%) | **Context**
*Bot*: Have you been hiking lately?
*User*: We have been especially with all the current issues, being outside is the only way to stay sane.
*Bot*: Do you sign up for the AllTrailsPro newsletter? They offer some great suggestions on trails to try
*User*: Yes i have actually been on that newsletter for years, they also have an amazing app that shows you what is around and how hard the hike is.
**Cosmo Response**
*Bot*: That sounds like a great app! i would love to try it out sometime.
**Zero-shot T5 with Cosmo**
*Entity:* Hiking
*Query:* What is the AllTrailsPro newsletter about?
*Problem:* The generated query was already answered previously so it's not interesting
**Finetuned T5 with Cosmo**
*Entity:* Hiking
*Query:* What are some popular hiking trails in the AllTrailsPro newsletter? |
| Query **Instruction Mismatch** (23.5%) | **Context**
*Bot*: I love Federer and Nadal when they paired up and played.
*User*: Yes that was great. Are you interested in becoming a tennis champion?
*Bot*: That would be a dream come true, but Tennis requires lots of physical effort.
*User*: It does. I have been training in Miami for years now. How long have you played?
*Bot*: I have had some lessons from Miami Tennis instructors and played for two years.
*User*: That is fantastic. I have played my whole life. My dream is to one day compete in Wimbledon
**Cosmo Response**
*Bot*: That would be amazing. I hope to one day compete in Wimbledon too.
**Zero-shot T5 with Cosmo**
*Entity:* Tennis
*Query:* How long have you been playing tennis?
*Problem:* The query is asking a question to the user
**Finetuned T5 with Cosmo**
*Entity:* Tennis
*Query:* What are the best tennis instructors in Miami? |
| Query **Others** (15.7%) | **Context**
*Bot*: Hello, welcome to Alexa social bot. What do you want to chat?
*User*: I like listening to NF because I like rap, and I appreciate the vulnerability in the lyrics
**Cosmo Response**
*Bot*: I'm glad you enjoy his music! i think that rap is a great genre to listen to, and it's good to see artists who are vulnerable in their lyrics.
**Zero-shot T5 with Cosmo**
*Entity:* Listen to NF
*Query:* What genre of music does NF like?
*Problem:* Query generated is factually wrong: NF is a singer, not a listener.
**Finetuned T5 with Cosmo**
*Entity:* NF
*Query:* What are some of NF's most vulnerable lyrics? |

Table 3: Qualitative examples of errors types in the query generation.

# Human Evaluation Guidelines

## Evaluation of Generated Query (200 examples)

## Instructions

You will be given a short human-bot conversation (ending with a user utterance) and a bot-generated search query to gather information for the next bot response. You need to first read the conversation and then evaluate the search query based on the following aspects described next.

*Note*: Current search systems can handle phrase-based and natural language queries. Hence, you should evaluate queries irrespective of how well formed they might be as long as they have all the important terms within them. For instance, the phrase-based query - "shoes for long distance marathon running" and the natural language query - "What are some good shoes for running long distances in marathons?" can be considered to be similar.

### Relevance

This evaluates for how relevant the search query is to the dialog. Ideally, the search query is expected to be about the topic of discussion in the last 2-3 turns. The results from the search query should be expected to keep the discussion coherent and not make it go off topic.

*Scoring Rubric:*
You will have to rate this on a likert scale of 1 to 5:
1: Completely Irrelevant
3: Vaguely Relevant
5: Highly Relevant

### Specificity

This evaluates for how specific the query is to the relevant topic being discussed. Queries can become more specific based on what fine-grained aspects of the topic the query explores.

For instance, below is an ordering of search queries based on specificity:

*Adidas running shoes for marathons > running shoes from Adidas > What are some good running shoes? > shoes*

*Scoring Rubric:*
You will have to rate this on a likert scale of 1 to 5:
1: Very generic
3: Somewhat specific
5: Highly specific

## Usefulness

This evaluates how useful the search query is for obtaining information to continue the conversation. This is based on the expectation of the usefulness of the search results corresponding to the given query.

*Note*: If you are unsure whether a query can be useful, you can pass it to google search and see whether the top-3 results can be useful.

*Scoring Rubric:*
You will have to rate this on a likert scale of 1 to 5:
1: definitely not useful
3: somewhat useful
5: definitely useful

## Interestingness

This evaluates whether the query can obtain information that can make the dialog very engaging. Such queries, while still keeping the discussion coherent i.e. being on-topic, have the potential to take the conversation to new and interesting directions.

*Note*: Interestingness is the highest bar for evaluation. A query that is irrelevant is not expected to be useful. A query that is relevant but very generic can still be somewhat useful (based on quality of ranking). However, a query that can provide interesting information is usually expected to be at least relevant, useful and somewhat specific.

*Scoring Rubric:*
You will have to rate this on a likert scale of 1 to 5:
1: very routine / uninteresting
3: somewhat interesting
5: definitely interesting

# Evaluation of Generated Responses (200 examples)

## Instructions

You will be given a short human-bot conversation (ending with a user utterance) and the next bot utterance. You need to first read the conversation and then evaluate the bot response on the following aspects described below.

## Coherence

This evaluates how well the response continues the dialog. Ideally, a coherent response is expected to be about the latest topic of discussion and show follow the logic flow of the dialog

*Scoring Rubric:*
You will have to rank the responses according to which one is more coherent given the dialog context.

## Engaging

A response is engaging if it has the potential to spark a conversation or further dialogue. Evaluation would be from the perspective of how likely it is that you would continue a conversation with the bot, given this response.

**Note**: Since the evaluation of level of engaging is based on the quality of the information in the response and not the actual response generator, you should ignore any follow-up questions from the bot that are present in the response. Responses with follow-up questions can appear more engaging than those without follow-up questions, hence we don't consider the follow-up questions to be fair to all query generation approaches.

*Scoring Rubric:*
You will have to rank the responses according to which one is more engaging.

## Informative

This evaluates for whether the bot response is knowledgeable and the main points of the response are still relevant to the conversation or topic. Informative responses involve the use of evidence, examples and provide enough novel content (something not already in the dialog history) to satisfy the user's curiosity.

*Scoring Rubric:*
You will have to rank the responses according to which one is more informative.

# GPT-4 Automatic Evaluation Prompt

## Query Scoring

We would like to request your evaluation of the performance of 5 systems, each of which generates a search query for obtaining relevant information to continue a conversation.

You will be given a short human-bot conversation (ending with a user utterance) and generated search query from each of the 5 systems, to gather information for the next bot response. You need to read the human-bot conversation and judge the overall quality of each of the search queries based on the relevance, specificity, usefulness and interestingness of the search query.

Human-Bot Conversation:

**<Context>**

System1 Query: <system 1>
System2 Query: <system 2>
......
System5 Query: <system 5>

For each of the system queries, you need to provide a single overall score in the range of 1-10, where a higher score indicates better overall performance. You should evaluate queries irrespective of how well formed they might be, as long as they have all the important terms within them. Note that interestingness is the highest bar for evaluation: A query that can provide interesting information for continuing a conversation is preferred, which such queries usually expected to be at least relevant, useful and somewhat specific. Please ensure that the order in which the queries were presented does not affect your judgment.

Evaluation (scores in the range of 1-10 ONLY)

**System1:**
**System2:**
**...**
**System5:**

## Query Preference

We would like to request your evaluation of the performance of 2 systems, each of which generates a search query for obtaining relevant information to continue a conversation.

You will be given a short human-bot conversation (ending with a user utterance) and generated search query from each of the 2 systems, to gather information for the next bot response. You need to read the human-bot conversation and judge the overall quality of each of the search queries based on the relevance, specificity, usefulness and interestingness of the search query.

Human-Bot Conversation:

**<Context>**

System1 Query: <system 1>
System2 Query: <system 2>

You should evaluate queries irrespective of how well formed they might be, as long as they have all the important terms within them. Note that interestingness is the highest bar for evaluation: A query that can provide interesting information for continuing a conversation is preferred, which such queries usually expected to be at least relevant, useful and somewhat specific. Please ensure that the order in which the queries were presented does not affect your judgment.

Evaluation - Which system query is better? (Just output system1 or system2)

**Output:**

## Response Scoring

We would like to request your evaluation of the performance of 2 systems, each of which generates a search query for obtaining relevant information to continue a conversation.

You will be given a short human-bot conversation (ending with a user utterance) and generated search query from each of the 2 systems, to gather information for the next bot response. You need to read the human-bot conversation and judge the overall quality of each of the search queries based on the relevance, specificity, usefulness and interestingness of the search query.

Human-Bot Conversation:

**<Context>**

Bot1 Response: <Response 1>
Bot2 Response: <Response 2>
......
Bot5 Response: <Response 5>

For each of the bot responses, you need to provide a single overall score in the range of 1-10, where a higher score indicates better overall performance. Please ensure that the order in which the responses were presented does not affect your judgment.

Evaluation (scores in the range of 1-10 ONLY)

**Bot1:**
**Bot2:**
**Bot3:**
**Bot4:**
**Bot5:**