# OpenReview forum: "Social Commonsense-Guided Search Query Generation for Open-Domain Knowledge-Powered Conversations"
_EMNLP/2023/Conference — EMNLP 2023 Findings_

### Official Review · Reviewer_XjaA · 2023-08-03

**Soundness:** 3

**Excitement:**

4: Strong: This paper deepens the understanding of some phenomenon or lowers the barriers to an existing research direction.

**Paper Topic And Main Contributions:**

This paper focuses on an unexplored conversational query generation problem setting that generates queries with ambiguous user intent. To tackle this problem, this paper leverages social commonsense reasoning to guide query generation. The main contribution of this paper is the query generation framework that consists of a finetuned topic identifier, a fixed commonsense-based response generation (i.e., Cosmo), and a finetuned instruction-based query generation.

**Questions For The Authors:**

Question A

In Line 190, can you explain the reason for reranking the top-3 search results? And what is the impact of providing all these search results for response generation?

Question B

In Table 1 (right), why do you omit the evaluations on ChatGPT results?

**Reasons To Accept:**

1. Providing an informative response to a passive user utterance can increase the engagingness of the chatbot. This paper first discusses the technique for generating a query when the user intent is unclear.

2. The experiment results are convincing with sufficient human and automatic evaluations. The human evaluation considers both query generation and response generation. The authors provide a good example of query generation human evaluation with detailed guidance.

**Reasons To Reject:**

1. Although the idea is good, the queries generated by this framework are still inferior to those generated by ChatGPT. This paper fails to state the benefits of using this framework to generate queries instead of utilizing ChatGPT directly.

2. This paper can be improved by more discussions about the generated query. For example, there is no discussion about why the zero-shot model with Cosmo receives the highest informativeness score in the human evaluation (in Table 1) and is even superior to the finetuned model.

3. The effectiveness of commonsense reasoning for a finetuned Flan-T5 remains unanswered. As stated in Line 178, the authors finetune Flan-T5 with ChatGPT results. Given that ChatGPT can produce queries with the highest evaluation scores, the compared baselines did not include the finetuned Flan-T5 without commonsense reasoning. This resulted in less robust ablation experiments.

**Reproducibility:**

4: Could mostly reproduce the results, but there may be some variation because of sample variance or minor variations in their interpretation of the protocol or method.

**Reviewer Confidence:**

4: Quite sure. I tried to check the important points carefully. It's unlikely, though conceivable, that I missed something that should affect my ratings.

**Typos Grammar Style And Presentation Improvements:**

The coherence score of "Ours (Finetuned)" in Table 1 is not the highest, but it is highlighted in bold.

---

> ### Author Rebuttal · Authors · 2023-08-29
>
> We would like to thank the reviewer for the valuable feedback and suggestions. Here, we address the reviewer’s concerns and questions.
>
> **Model does not outperform ChatGPT**: Our model may not surpass ChatGPT in performance; however, we would like to highlight that we employ a considerably more compact model, Flan T5 Large with 770M parameters, for generating queries. This is complemented by the 3B parameter COSMO model that supplies commonsense knowledge. In real-world dialog systems where swift response times are crucial, it is advantageous to utilize a smaller model for query generation. Larger models like ChatGPT are generally reserved for the final stages of response creation.
>
> **Zero-shot model with Cosmo seems to be slightly better than finetuned one at informativeness of responses**: When directly evaluated for query generation performance, we see that the finetuned model considerably outperforms the zero-shot model. Yet, when assessed based on the final response quality—derived from search results obtained using the generated queries—the gap between the finetuned and zero-shot models narrows. This discrepancy arises because the final response generation includes web search, ranking, and prompting ChatGPT to generate a response with those search results. Even if the query quality differs, the eventual response might not reflect that due to errors in these stages. For instance, even with a relevant query, the web search and ranking system might occasionally pick unrelated content. Moreover, ChatGPT tends to (a) compensate heavily for unrelated content, ensuring a coherent and relevant final response and (b) sometimes overlooks crucial content from the search, omitting vital elements.Therefore, even though the quality of the final response can provide a rough measure of the search query's effectiveness, it's not devoid of potential biases. We will incorporate this discussion into our final draft.
>
> **Finetuned Flan-T5 without commonsense reasoning not compared**: We thank the reviewer for this excellent suggestion. We will add this comparison, with both human and automatic evaluation, into the final version of the paper.
>
> **Question A**: Each search result is derived from the unprocessed content of a webpage, which might be extensive and filled with unnecessary details. To address this, we segment the webpage into passages and employ a re-ranker to identify the most pertinent section within the page.
>
> **Question B**: In Table 1 (left), ChatGPT mainly serves as an upper bound for model performance on search query generation. Given constraints in annotator time and budget, Table 1 (right) focuses on evaluating *ChatGPT (No Query)* as a more insightful baseline for final response, rather than assessing the response generated based on the search results obtained by a ChatGPT-generated query.

---

### Official Review · Reviewer_Vk7b · 2023-08-03

**Soundness:** 3

**Excitement:**

3: Ambivalent: It has merits (e.g., it reports state-of-the-art results, the idea is nice), but there are key weaknesses (e.g., it describes incremental work), and it can significantly benefit from another round of revision. However, I won't object to accepting it if my co-reviewers champion it.

**Paper Topic And Main Contributions:**

This paper tackles the problem of generating queries targeting passive users using social commonsense.
The paper contributes with a pipeline composed of 3 main parts: 1) a fine-grained topic tracking method using FLAN-T5 to extract the current topic of the conversation; 2) a commonsense model using Cosmo that receives the topic and generates a latent commonsense directive to guide the query generation procedure; and 3) an instruction-driven query generation method, also using FLAN-T5, prompted to convert the response into a search query to be used in a search engine.
To evaluate, the authors use the Wizard of Internet dataset and evaluated with both human evaluation, and automatic evaluation using GPT-4.
The results showed that the proposed method was better at generating queries and generating a final response in both types of metrics given the considered baselines. The results also showed that incorporating commonsense using Cosmo led to better performance, and that further fine-tuning the model on ChatGPT answers also helps.

**Questions For The Authors:**

A) What type of decoding strategy and parameters were used in each component of the framework?

**Reasons To Accept:**

1) The results achieved are encouraging, showing improvements over the considered baselines.
2) Tries to solve a practical problem of handling passive user interactions.
3) The evaluation considered both human-judgments and model-based judgments, and the authors also performed an additional error analysis.

**Reasons To Reject:**

1) I would have liked to see a more complete discussion about how incorporating the social commonsense influences the query generation procedure, besides the results of not using Cosmo.
2) There is some under-specification in the parameters used to generate the answers of the different models, hindering reproducibility of the work.

**Reproducibility:**

4: Could mostly reproduce the results, but there may be some variation because of sample variance or minor variations in their interpretation of the protocol or method.

**Reviewer Confidence:**

4: Quite sure. I tried to check the important points carefully. It's unlikely, though conceivable, that I missed something that should affect my ratings.

**Typos Grammar Style And Presentation Improvements:**

The paper is well written and easy to follow.

When mentioning ChatGPT, please specify which version you are using and if you used Open AI’s API.

Regarding the instructions given to the annotators, it would probably be good to have them in another format that would allow for easier reading instead of just being appended to the document.

---

> ### Author Rebuttal · Authors · 2023-08-29
>
> We woud like to thank the reviewer for the valuable feedback. Here, we address the reviewer’s concerns.
>
> **More discussion on how incorporating social commonsense helps query generation**: We utilize the Flan T5 without Cosmo system as the main comparison point to demonstrate the advantages of integrating commonsense knowledge. Additionally, in Section 3.4, we discuss how the integration of commonsense knowledge affects both larger and smaller query generation models. The introduction section, from lines 50-57, outlines the rationale for incorporating social commonsense. Further, lines 78-86, along with Figure 1, offer a detailed example illustrating the impact of social commonsense on search query formulation.
>
>
> **Decoding Strategy and Parameters not provided**: For both search query generation and final response generation,we employ the Top-P sampling method with P set at 0.9 and a temperature of 0.7. We set max tokens to 40 and 100 for search query generation and final response generation respectively. For ChatGPT, we use the *gpt-3.5-turbo-0301* version. These details will be added into the final version of the paper.

---

### Official Review · Reviewer_TgDw · 2023-08-05

**Soundness:** 3

**Excitement:**

3: Ambivalent: It has merits (e.g., it reports state-of-the-art results, the idea is nice), but there are key weaknesses (e.g., it describes incremental work), and it can significantly benefit from another round of revision. However, I won't object to accepting it if my co-reviewers champion it.

**Paper Topic And Main Contributions:**

This paper focuses on the problem of increasing user engagement in the interactions with an open domain dialog system by leveraging common sense reasoning techniques (applied to the conversation content). The main contribution of the paper is the overall system proposed, together with the experimental results demonstrating the effectiveness of the system.

**Questions For The Authors:**

It is not clear to me what the "ChatGPT" results in query generation indicate (the last line of Table 1). The scores are higher than all other entries (both baselines and proposed systems) - is this intended to be an oracle system of some sort? If so, please indicate accordingly.

I am also curious about the statistical significance. Can you clarify whether the comparison is done with both baseline methods, or just the worse of the two baselines?

I am somewhat puzzled by the marking of statistical significance in the final response section of table 1. Only GPT4-Zero Shot cell is marked as significantly improving over the baselines; but GPT4-Fine Tuned has a higher result, so I would expect it to also be significant. Can you double check the significance markings in that section?

I would like to see the distribution of errors after fine tuning, too - can you include that in section 3.4 at the end of the Error Categorization paragraph? I am curious if any particular class of errors is helped by the fine tuning process.

**Reasons To Accept:**

I find the paper generally well written, clear, and easy to follow. The system proposed appears to be effective, based on the experimental results shown in table 1. Results show statistical significance over the baseline for query generation (other than the last line, for which I have a question, below).

**Reasons To Reject:**

While results in the Final Response section are promising, the paper appears to overemphasize the achievements; there is little statistically significant improvement in this section (with one question about this included below); the main achievement appears to be in matching the coherence obtained from the no-search configuration (first line) while also matching (or perhaps slightly improving upon) the engagement and informativeness of Flan T5 w/o Cosmo. This is fine but the paper as written claims more, which I find misleading.

**Reproducibility:**

4: Could mostly reproduce the results, but there may be some variation because of sample variance or minor variations in their interpretation of the protocol or method.

**Reviewer Confidence:**

4: Quite sure. I tried to check the important points carefully. It's unlikely, though conceivable, that I missed something that should affect my ratings.

---

> ### Author Rebuttal · Authors · 2023-08-29
>
> We thank the reviewer for the valuable feedback and comments. Here, we address the reviewer’s concerns and answer the questions they have raised.
>
> **Minor improvements in the final response**: It's important to clarify that our primary focus in this study is on search query generation.  However, producing the final response introduces steps like web search, ranking, and prompting ChatGPT to generate a response using the search results. Due to these additional components, the following errors arise:
>
> - Despite a relevant query, the web search and ranking process can occasionally select irrelevant content.
> - ChatGPT often (a) adjusts excessively for irrelevant content to still create a coherent and relevant final response and (b) does not fully utilize pertinent search content, sometimes missing key elements -  this is evidenced by an average ROUGE-L overlap of only 0.25 between the generated response and the search content.
>
> Thus, even when there is a significant difference in the quality of generated queries  (supported by both human and machine evaluations), the differences do diminish in the final response due to errors introduced by the steps mentioned above.
>
> **What do “ChatGPT” query generation numbers correspond to?**: We consider ChatGPT as the state-of-the-art model for search query generation, and show numbers for evaluation of ChatGPT generated queries as a measure of upper-bound for model performance. While these numbers serve as a dynamic standard for upper-bound of model performance, we anticipate even better results with GPT-4, especially considering our experiments utilized gpt-3.5-turbo-0301.
>
> **Clarification of statistical significance results**: To determine statistical significance, we compared the zero-shot version of our model, denoted as "Ours (Zero-shot)" in Table 1, with all baseline methods. Meanwhile, the finetuned version of our model, labeled as "Ours (Finetuned)" in Table 1, is compared against our zero-shot model. These details will be more explicitly outlined in the final version.
>
> **Error Distribution After Finetuning**: After finetuning, of the examples that continue to score low, the breakdown is as follows: 19% result from *Incorrect Topic*, 41% from *Trivial Queries*, 5% from *Query Instruction Mismatch*, and 35% from other unrelated queries. Notably, finetuning effectively reduces 67% of *Trivial Query* errors, 75% of *Incorrect Topic* errors, and over 90% of *Query Instruction Mismatch* errors. This suggests that finetuning enhances topic tracking capabilities (resulting in fewer *Incorrect Topic* errors) and ensures better adherence to search query generation instructions (leading to a decrease in *Query Instruction Mismatch* errors).

---

### Meta-Review · Area_Chair_SwVN · 2023-09-21

**Recommendation:** 3

**Metareview:**

This paper focuses on an unexplored conversational query generation problem setting that generates queries with ambiguous user intent. To tackle this problem, this paper leverages social commonsense reasoning to guide query generation. The main contribution of this paper is the query generation framework that consists of a finetuned topic identifier, a fixed commonsense-based response generation (i.e., Cosmo), and a finetuned instruction-based query generation.

The problem is practical and the results achieved are encouraging. Nevertheless, reviewers have the following concerns:
1. Minor improvements in the final response. While the authors provide their response, I do not think this concern has been fully addressed. From the response, it seems that high-quality queries are not that important.
2. The paper fails to state the benefits of using the proposed framework to generate queries instead of utilizing ChatGPT directly. Besides, there are no efficiency analyses.
3. Finetuned Flan-T5 without commonsense reasoning not compared.

---

### Decision · Program_Chairs · 2023-10-07

**Decision:**

Accept-Findings

**Comment:**

This paper focuses on an unexplored conversational query generation problem setting that generates queries with ambiguous user intent. To tackle this problem, this paper leverages social commonsense reasoning to guide query generation. The main contribution of this paper is the query generation framework that consists of a finetuned topic identifier, a fixed commonsense-based response generation (i.e., Cosmo), and a finetuned instruction-based query generation.

The problem is practical and the results achieved are encouraging. Nevertheless, reviewers have the following concerns:
1. Minor improvements in the final response. While the authors provide their response, I do not think this concern has been fully addressed. From the response, it seems that high-quality queries are not that important.
2. The paper fails to state the benefits of using the proposed framework to generate queries instead of utilizing ChatGPT directly. Besides, there are no efficiency analyses.
3. Finetuned Flan-T5 without commonsense reasoning not compared.